# Spheno-Orbital Meningioma and Vision Impairment—Case Report and Review of the Literature

**DOI:** 10.3390/jcm12010074

**Published:** 2022-12-22

**Authors:** Joanna Wierzbowska, Arkadiusz Zegadło, Michał Patyk, Marek Rękas

**Affiliations:** 1Department of Ophthalmology, Military Institute of Medicine—National Research Institute, 04-141 Warsaw, Poland; 2Department of Radiology, Military Institute of Medicine—National Research Institute, 04-141 Warsaw, Poland

**Keywords:** spheno-orbital meningioma, optic neuropathy, vision impairment, vision loss

## Abstract

(1) Background: Spheno-orbital meningioma (SOM) is a very rare subtype of meningioma which arises from the sphenoid ridge with an orbital extension. It exhibits intraosseous tumor growth with hyperostosis and a widespread soft-tissue growth at the dura. The intra-orbital invasion results in painless proptosis and slowly progressing visual impairment. (2) Methods: We present a case of a 46-year-old woman with SOM and compressive optic nerve neuropathy related to it. Her corrected distance visual acuity (CDVA) was decreased to 20/100, she had extensive visual field (VF) scotoma, dyschromatopsia, impaired pattern-reversal visual-evoked potential (PVEP), and decreased thicknesses of the retinal nerve fiber layer (RNFL) and ganglion cell complex (GCC), measured with the swept-source optical coherence tomography (SS-OCT), and a pale optic nerve disc in her left eye. Brain CT and MRI showed a lesion at the base of the anterior cranial fossa, involving the sphenoid wing and orbit. Pterional craniotomy and a partial removal of the tumor at the base of the skull and in the left orbit with the resection of the lesional dura mater and bony defect reconstruction were performed. (3) Results: The histological examination revealed meningothelial meningioma (WHO G1). Decreased CDVA and VF defects completely recovered, and the color vision score and PVEP improved following the surgery, but RNFL and GCC remained impaired. No tumor recurrence was observed at a follow-up of 78 months. (4) Conclusions: Optic nerve dysfunction has the capacity to improve once the compression has been relieved despite the presence of the structural features of optic nerve atrophy.

## 1. Introduction

Spheno-orbital meningioma (SOM) is a rare meningioma arising from the sphenoid wing with a periorbital extension. It accounts for 2–9% of all intracranial meningiomas [1,2,3,4,5]. These complex, slow-growing tumors have characteristic morphological and clinical features. They comprise two components: an intraosseous growth with secondary associated hyperostosis and an intradural, soft-tissue component. The bony tumor growth typically involves the sphenoid ridge as well as the lateral and superior orbital walls, and might involve the superior orbital fissure, optic canal, and anterior clinoid process. The dural growth is usually widespread and carpet-like, including the basal sphenoid wing, cavernous sinus, and temporal convexity [5,6]. Despite its benign histopathological features, they are aggressive in behavior in the long clinical course because they can infiltrate the optic canal and exert a mass effect. The clinical features of SOM result from these intraosseous, intradural, and intra-orbital lesions and include a triad of symptoms, in descending order of frequency: proptosis, visual impairment, and ocular motility defects [4]. Visual impairment is a result of the optic canal invasion, optic nerve (ON) compression, periorbital tissue infiltration, or orbital apex invasion by the tumor [7] and occurs in 40–60% of cases [6,8]. The treatment for symptomatic or progressing tumors involves their surgical removal. The goals of SOM surgery include the restoration of visual function and the reduction of proptosis, rather than total oncological resection.

In the vast majority of reported cases concerning spheno-orbital meningiomas (SOMs), visual impairment was characterized by a corrected distance visual acuity (CDVA) change only. We report a case of a 46-year-old woman with SOM, who developed severe compressive optic neuropathy (CON). To assess the function and structure of the optic nerve, we examined CDVA, visual field (VF) for the mean deviation (MD) and pattern standard deviation (PSD), a color test, pattern-reversal visual-evoked potential (PVEP) for P100 latency and amplitude, and peripapillary retinal nerve fiber layer (RNFL) and ganglion cell complex (GCC) thicknesses. To the best of our knowledge, no case reports are available that address such a multimodal approach to evaluate both the function and structure of the ON in a seven-year course of SOM.

We also present a literature review of case series of SOMs to summarize both visual impairment caused by this disease and visual outcomes following tumor resection.

## 2. Case Report

A 46-year-old woman was referred to the Clinic of Ophthalmology, Military Institute of Medicine in Warsaw for the diagnosis of the lesion in the left orbit. The patient was referred by another ophthalmologist with the suspicion of craniofacial fibrous dysplasia, which was based on orbital B-scan ultrasonography findings. She suffered from the progressive deterioration of vision in her left eye and slowly progressive proptosis on the left side for over two years. She also complained of slight pain of the left parietal region, upper eyelid edema, and epiphora for 4 months. She had no diplopia. She was also treated for iron deficiency anemia. No signs of neurofibromatosis were noted.

Ocular examination showed CDVA to be 1.0 (20/20) in the right eye (RE) and 0.2 (20/100) in the left eye (LE), assessed with the Snellen E letter chart from the distance of six meters. There was 3 mm of proptosis of the LE assessed by Hertel exophthalmometry, but eye movements were normal. A relative afferent pupillary defect (RAPD) was present in the LE, and the color tests score (by the Ishihara color plate test) was decreased to 7/14. Intraocular pressure measured by applanation tonometry was 16 mmHg in the RE and 20 mmHg in the LE. Slight upper eyelid edema of the LE was also noted. Dilated fundus examination revealed mild pallor of the left optic disc. Humphrey standard automated perimetry showed a temporal and inferior nasal VF defect of the LE (MD −18.78 dB, PSD 9.39 dB). The VF of the RE was normal. PVEP showed that P100 latencies were similar on the left and right sides (110 ms vs. 102 ms). However, the amplitude of P100 was decreased on the left side (3.4 µV vs. 17.1 µV). The mean peripapillary retinal nerve fiber layer (RNFL) thickness measured with swept-source optical coherence tomography (SS-OCT) was slightly decreased in the LE compared to the RE (101 µm vs. 110 µm). The lowest RNFL thickness in the LE was reported in the temporal quadrant (38 µm in the LE vs. 76 µm in the RE). Additionally, the macular ganglion cell complex (GCC) thickness, which comprises the three innermost layers of the retina: the RNFL, the ganglion cell layer (GCL), and the inner plexiform layer (IPL), was decreased in the LE as compared to the RE (84 µm vs. 99 µm).

Computed tomography (CT) performed at our institution revealed the hyperostosis of the left greater sphenoid wing and lateral orbital involvement by a tumor, with optic canal narrowing and lateral and superior extraocular muscle modeling. Gadolinium-enhanced magnetic resonance imaging (MRI) revealed an enhancing tumor along the greater wing of the left sphenoid bone, dural involvement, and intra-orbital tumor infiltration with optic nerve compression (Figure 1).

The patient was referred to the Clinic of Neurosurgery of the Medical University in Bialystok for spheno-orbital lesion removal. Pterional craniotomy and a partial removal of the tumor at the base of the skull and in the left orbit with the resection of the involved dura mater were performed. Simultaneously, the dural and bony defects were reconstructed with collagen matrices, fibrin sealant patches, and a titanium mesh implant.

Histopathological examination revealed meningothelial meningioma (WHO G1) with the features of infiltration into the dura and bones. There were no major postoperative complications.

Three weeks postoperatively, the CDVA of the LE improved to 0.8 (20/25) (Snellen). Subsequent follow-up visits revealed the resolution of VF scotoma and the normalization of the CDVA to 1.0 (20/20) and Humphrey VF indices (MD −1.48 dB, PSD −2.01 dB). The color vision score improved from 7/14 to 12/14. Simultaneously, a reduction of the proptosis of the LE was observed. Six months after the surgery, we found a slight increase in the amplitude of PVEP and latency delay. The P100 latencies of PVEP were 104.5 ms on the right side and 136 ms on the left side. During the 6-month follow-up, SS-OCT RNFL thickness in the affected eye continued to decrease in comparison with the preoperative values (85 µm vs. 101 µm), but then it did not change until the end of the 78-month observation (Figure 2).

Follow-up MRI (Figure 3) and CT (Figure 4) scans showed a contrast enhancement area with tumor residue but with no sign of progression. There was no tumor recurrence or CDVA and VF deterioration at the follow-up of 78 months.

## 3. Systematic Review

A systematic review was performed according to the PRISMA (Preferred Reporting Items for Systematic Reviews and Meta-Analyses) criteria [9]. The PubMed, Medline, and Scopus databases were searched for relevant literature on 25 October 2022. The keywords and their derivatives or synonyms were combined in each database as follows: “spheno-orbital meningioma” and “vision impairment” or “vision deficits” or “vision loss” or “optic neuropathy”. Spheno-orbital meningioma was defined as an intraosseous meningioma with extensive hyperostosis, which involves the sphenoid wing and the orbit. The inclusion criteria were: (i) patients with spheno-orbital meningioma with visual symptoms who underwent complete or incomplete surgical resection, (ii) papers published in the English language, and (iii) papers published in a peer-reviewed journal between 2000 and 2022. The exclusion criteria were: (i) non-hyperostotic spheno-orbital meningioma, (ii) sphenoid wing meningioma with no orbital extension, (iii) meningiomas with isolated intra-orbital location, (iv) conference proceedings or books, and (v) single-case reports, reviews, letters to the editors, radiological studies, or only abstracts. In case of several articles describing overlapping cohorts, the most recent article was included.

After the removal of duplicates, the titles and abstracts of all articles were screened, and potentially relevant articles were included into the full-text review by two independent reviewers (J.W. and A.Z.). If disagreements occurred among the investigators, these were discussed, and the senior investigator (J.W.) made the final decision. A total of 33 case series met the full criteria for inclusion. Clinical data on age, sex, pre- and post-operative visual deficits, the length of follow-up, and the rate of tumor recurrence were reviewed.

The PRISMA flowchart of literature selection in this review is illustrated in Figure 5.

## 4. Results

The results of the included studies are summarized in Table 1. They include the demographic features of SOMs as well as data on preoperative visual impairment, postoperative visual outcomes, the length of follow-up, and the rate of tumor recurrence.

Through the literature search and screening, 33 articles on SOM and visual impairment were included in this review (Table 1). All studies were retrospective, based on medical record chart review. Due to the low incidence of SOMs and their slow growth, the included series describe both smaller and larger patient series, often covering two or three decades. A total of 1247 patients with SOMs were analyzed.

Visual impairment was defined mostly as a reduced CDVA assessed by the Snellen chart. The clinical outcome was assessed by comparing preoperative and postoperative CDVA. No cut-off for improvement was set by the vast majority of authors. Other clinical evidence of ON dysfunction, including computerized VF defects, was used as an additional tool only by fewer than half of the authors [4,6,12,13,14,17,18,23,24,25,26,29,32,33,35]. A multimodal approach to the measurement and monitoring of optic nerve function additionally including color vision testing (with Ishihara plates) and pupillary reaction testing was only performed in one case series [18]. If a multimodal approach was applied, only one of the above clinical findings had to be present to confirm visual impairment or optic nerve dysfunction.

The extent of tumor resection was categorizIed in the reviewed literature according to the Simpson grading scale [36] as total macroscopic (TTR, Simpson grade I), near-total (NTR, Simpson grade II), subtotal (STR, Simpson grade III), and partial tumor resection (PTR, Simpson grade IV). TTR and NTR were considered when surgery left no visible residual tumor on the follow-up MRI and CT scans, whereas STR and PTR were defined as a residual tumor that was left within the cavernous sinus or intradurally, respectively. Some authors used the term “gross total resection” (GTR) as equivalent to the terms TTR (Simpson grade I) and NTR (Simpson grade II), and others see GTR as equivalent to TTR only.

In the study by Mariniello et al. [10], 80 patients operated on for SOMs were retrospectively reviewed. Visual impairment was present in 47 patients (59%). The tumor location within the orbit was lateral (type I) in 20 patients (25%), medial (type II) in 13 (16%), at the orbital apex (type III) in 30 (38%), and diffuse (type IV) in 17 (21%). The involvement of the optic canal was found in 59 patients (79%). Total tumor resection was obtained in 100% of type I tumors, in 77% of type II, in 63% of type III, and in 18% of type IV tumors. Postoperatively, visual improvement or deterioration were noted in 51% and 15% of patients, respectively.

A study by dos Santos et al. [11], including 40 patients with SOMs, showed decreased CDVA in 26 patients (65%). The standard surgery involved the performance of pterional craniotomy with superolateral orbitotomy. GTR was achieved in 65% of the procedures. Postoperatively, 41% and 37% of the patients improved and maintained CDVA, respectively.

Locatelli et al. [12] identified 11 patients with visual impairment among 35 patients treated surgically for SOMs. The patients were operated through craniotomic (49%), endoscopic superior eyelid (37%), and combined cranio-endoscopic (14%) approaches with neuro-navigation. GTR was obtained in 76% of patients operated with the craniotomic approach against 46% of patients operated with the endoscopic approach. The late outcomes showed that CDVA improved in 7 out of 11 patients with preoperative visual deficits, independently from the approach used. None experienced vision deterioration.

Dalle Ore et al. [13] conducted a retrospective review of 54 patients undergoing the resection of SOMs. Twenty-eight patients (52%) manifested decreased preoperative CDVA and VF deficits. GTR of the intracranial and orbital tumor was achieved in 43% and 27%, respectively. Postoperative vision was improved or stable in 97% of patients with preoperative visual impairment.

The study by Najafabadi et al. [14] focused on the predictors of postoperative visual outcomes in 10 patients with SOMs and accompanying CDVA and VF deficits. Linear regression analysis showed that poorer preoperative CDVA (β = −0.49, 95%CI −0.21 to −0.77, *p* = 0.002) and the diagnosis of multiple meningioma (β = −0.14, 95%CI −0.26 to −0.02, *p* = 0.021) were the predictors for poorer long-term CDVA. A higher diameter of hyperostosis (β = 0.39, 95%CI −0.67 to −0.12, *p* = 0.009) and a lower extent of tumor resection (β = 3.71, 95%CI −6.63 to −0.78, *p* = 0.017) were the predictors for poorer long-term visual field outcomes.

According to Menon et al. [15], 14 patients with preoperative visual impairment were identified among 17 patients surgically treated for SOMs. Extensive orbital wall decompression and optic canal deroofing were routinely performed. GTR was obtained in 12% of patients. Postoperatively, vision improved in 4 patients (29%), remained stable in 8 (57%), and deteriorated in 2 patients (’4%).

A case series by Samadian et al. [16], including 57 patients with SOMs undergoing surgery with the frontotemporal approach, showed that 16 patients (28%) experienced CDVA decline and VF deficits before surgery. GTR was achieved in 84% of the patients. Optic canal unroofing was performed in 43% of the patients. Seven patients showed CDVA improvement, and one patient showed CDVA deterioration following the surgery.

Pace et al. [17] conducted a retrospective chart review of 20 patients with SOMs who underwent frontotemporal and orbitozygomatic craniotomy with the removal of the whole bone affected by the tumor. The surgery also involved the unroofing of the optic canal, removal of the periorbital component, followed by orbital reconstruction. GTR was accomplished in 75% of the patients. Nine out of eleven patients (55%) with decreased preoperative CDVA experienced visual improvement.

As regards a study by Young et al. [18], 17 (71%) out of 24 cases with SOMs demonstrated visual impairment, defined as decreased CDVA and/or the presence of RAPD and/or reduced color vision and/or VF defect. Frontotemporal craniotomy and lateral orbital decompression were performed. A lateral 180-degree decompression of the optic canal was performed in majority of cases. Three months postoperatively, vision improved in 12, remained unchanged in 7, and deteriorated in 5 patients. Of the 12 cases who experienced visual improvement, CDVA improved in 4, RAPD resolved in 5, color vision improved in 4, and VF normalized in 7 patients. Visual deterioration was suffered by 5 patients, with aggressive tumor recurrence being the cause in one and postoperative optic nerve neuropathy in the remaining four patients.

Nagahama et al. [19] reported a surgical strategy of aggressive tumor resection with the opening of the optic canal and described the long-term outcomes of surgery for SOMs. The mean follow-up was 74 months (range: from 10 to 262 months). Three out of twelve patients with SOMs showed visual disturbances preoperatively and two of them experienced further visual worsening following radical tumor resection.

A large study by Terrier et al. [5] including 130 consecutive patients undergoing surgery for SOMs during a 20-year period revealed preoperative visual impairment in 49 patients (38%). Simpson grade I and II removal were achieved in 75% of the patients. After surgery, vision improved in 22 of 49 patients (45%), remained stable in 19 (39%), and deteriorated in 8 patients (16%). Four patients (8%) became blind. There was no association between the severity of preoperative visual impairment and the postoperative visual outcomes. Additionally, the duration of symptoms and the extent of surgical procedure, including dural or complete bone excision, periorbital excision, and the decompression of the optic canal, did not predict a significant vision improvement.

Gonen et al. [2] proposed an intraoperative decision-making algorithm for SOMs based on the surgical treatment of 27 patients. It included: (1) the extracranial stage (frontotemporal approach), (2) the extradural stage (clinoidectomy in cases of tumor invasion), (3) the intradural stage (resection of intradural component), (4) the intra-orbital stage (selective periorbital opening), and (5) duraplasty and orbital reconstruction. Surgery contributed to visual improvement in 8 out of 10 patients who demonstrated a preoperative vision decline.

Freeman et al. [3] conducted a retrospective review of 25 patients who underwent SOM resection followed by fractionated radiation at the time of recurrence (8 patients). Simpson grades I and II resection were achieved in 68% of patients. The mean follow-up time was 45 months. As regards the late outcomes, CDVA improved in 4, was stable in 14, and deteriorated in 1 out of 19 patients who presented with decreased CDVA preoperatively.

A study by Leroy et al. [20] compared postoperative visual acuity evolution between internal and external SOM varieties. They classified 40 cases as the internal SOM variety when the inner third of the sphenoid wing, optic canal, anterior clinoid process, or cavernous sinus were involved. The optic canal was involved in 24 patients (57%) in the group. Another 30 cases had the lesion described as the external SOM variety when at least 1 of the following structures exhibited invasion: the pterion, the eternal third of the sphenoid wing, or the external part of the orbit. Complete resection was obtained in 12% of internal variety patients compared to 61% of external variety patients. Postoperatively, no significant difference was noted between either SOM variety regarding CDVA changes, although decreased CDVA was reported more commonly in the internal variety group (17% vs. 4%, *p* = 0.13).

Bowers et al. [21] reviewed the outcomes of an aggressive surgical approach to the removal of SOMs in 33 patients. A total of 17 patients had decreased CDVA preoperatively, with 15 improving after surgery and none experiencing CDVA deterioration.

A large study by Amirjamshidi et al. [22], including 88 patients with SOM, evaluated the impact of preoperative variables upon different outcome measures. Decreased CDVA was present in 65 (75%) patients, including no light perception in 16 patients and light perception up to 0.2 in 3 cases. Twelve patients underwent standard pterional craniotomy and another seventy-six patients had lateral miniorbitotomy. Postoperatively, vision improved in 39 patients (60%) and deteriorated in 13 patients (20%). Three of the patients with no preoperative light perception achieved light perception following the surgery. In turn, 2 out of 47 patients with CDVA 0.5–1.0 became blind after the operation. The technique of the surgery was not associated with any postoperative outcomes.

Talacchi et al. [23] conducted a retrospective chart review of 47 patients with SOMs who underwent standard craniotomy without orbital reconstruction. GTR was accomplished in 24 cases (51%). At the 4- to 6-month assessment, 10 out of the 24 patients with decreased preoperative CDVA experienced visual improvement and 3 showed further CDVA deterioration.

A study by Berhouma et al. [24] evaluated the efficacy and safety profile of minimally invasive endoscopic endonasal optic nerve and orbital apex decompression for optic neuropathy in four patients with SOMs. The surgery resulted in visual improvement in three patients. However, one patient with optic nerve atrophy continued to worsen.

Boari et al. [25] reviewed the outcomes of combined surgical–radio-surgical treatment of SOMs in 40 patients. The primary procedure involved frontotemporal craniotomy and the secondary procedure, in case of subtotal resection, involved Gamma Knife radiosurgery on the residual tumor. Preoperatively, 33 patients manifested decreased CDVA. Vision improved in 27 (67%) of them after the treatment.

Marcus et al. [26] conducted a review of 19 patients with SOMs who underwent image-guided tumor resection with lateral orbital decompression. In 10 out of 11 patients presenting preoperative visual impairment, CDVA improved or stabilized after the treatment.

The purpose of a study by Nochez et al. [7] was to assess the radiological and perioperative predictive factors for CDVA evolution and VF outcomes based on 37 patients with SOMs. A total of 22 patients manifested decreased CDVA preoperatively. After the surgery, 12 of them improved, 5 stabilized, and 5 deteriorated. The extension to the periorbita was found to be a negative factor for the CDVA improvement. The invasion of the optic canal and the presence of a soft-tissue intracranial component were the predictors for severe postoperative VF defects.

Saeed et al. [27] evaluated the clinical outcomes following different surgical approaches (pterional craniotomy alone or combined with orbitozygomatic resection, as well as extended lateral orbitotomy alone) in 66 patients with SOMs. Progressive visual impairment was manifested by 51 patients preoperatively. After the surgery, 20 patients (39%) had their CDVA improved and 6 (12.5%) developed further visual decline. There was no significant difference (*p* = 0.195) in postoperative CDVA between different surgical approaches.

Oya et al. [28] conducted a retrospective review of 39 patients who underwent an aggressive resection of SOM. Postoperatively, the vision of 14 (67%) out of 21 patients with decreased preoperative CDVA improved. Multiple regression analysis adjusting for tumor characteristics revealed that severe sphenoid bone hypertrophy was an independent favorable factor for a better visual outcome after surgery (adjusted odds ratio 0.08, *p* = 0.035). Preoperative severe CDVA decline was an independent risk factor for postoperative CDVA improvement (*p* = 0.009).

Luetjents et al. [29] reported on the results of the surgical management of three patients with bilateral SOMs and an accompanying bilateral progressive loss of vision. Surgery was performed in two stages, primarily treating the most affected side. Vision improved (from the perception of movement, −0.3 to 0.2–0.6) in all patients, although transient visual deterioration was observed in the early postoperative period in one case.

Honig et al. [30] conducted a review of 30 patients with SOMs who underwent radical (33% of patients) and partial (67% of patients) tumor resection. Preoperative decreased CDVA and VF defects were noted in 22 and 12 patients, respectively. After the surgery, CDVA improved in 15 patients, remained the same in 5, and worsened in 2 patients.

A large case series by Mirone et al. [31], including 71 patients with SOM, evaluated clinical outcomes following tumor surgery based on a long-term follow-up (the mean of 77 months). Decreased CDVA was reported in 41 patients (58%). GTR (Simpson grades I and II) and optic canal decompression was performed in the vast majority of patients (83% and 75%, respectively). After the surgery, vision improved in 30 patients (73%), remained stable in 8 (19.5%), and deteriorated in 3 patients (7%) with preoperative CDVA below 0.2.

Heufelder et al. [32] reviewed the outcomes of SOM surgery in 21 patients. One out of seven patients with preoperative VF defects demonstrated persistent VF scotoma and two patients with decreased preoperative CDVA experienced blindness after surgery.

A study by Cannon et al. [33], including 12 patients with SOMs undergoing tumor resection with the pterional approach and lateral orbital wall decompression, showed that vision improved in 2 patients and deteriorated to blindness in 3 patients after the surgery.

A case series by Ringel et al. [6], including 63 patients with SOMs undergoing tumor resection, showed that 28 (47%) and 20 (32%) patients experienced preoperative CDVA decline and VF defects, respectively. The surgery improved CDVA in 64% of the patients, while VF improved in 58% of the patients. Two patients showed CDVA and VF deterioration following surgery.

Bikmaz et al. [1] conducted a review of 17 patients with SOMs who underwent radical (82% of patients) tumor resection. Preoperative progressive vision loss was noted in 10 patients (59%). Postoperatively, CDVA improved in 7 patients, and none continued to worsen.

Shrivastava et al. [4] evaluated clinical outcomes after a standardized surgical approach with the drilling of the optic canal and cranioorbital reconstruction in 25 patients with SOMs. As regards 20 patients with decreased preoperative CDVA, 7 patients experienced CDVA improvement, and none developed further visual decline after the surgery. The surgery contributed to VF improvement in 8 out of 9 patients who demonstrated VF defects preoperatively.

A series of 16 patients underwent a radical surgical resection of SOMs and they were reviewed by Sandalcioglu et al. [34]. CDVA improved or stabilized after the treatment in 6 out of 7 patients presenting preoperative progressive visual impairment. A long-term follow-up (up to 188 months) was conducted. It was found that four out of nine patients who experienced a recurrence of the tumor showed progressive visual loss, which remained stable after the second surgical procedure.

De Jesús et al. [35] conducted a review of six patients with SOMs who underwent tumor resection with orbital decompression. The surgery improved vision in all three patients who had preoperative visual defects.

## 5. Discussion

This study reviewed visual impairment in patients with SOMs before and after surgical treatment.

SOMs are defined as secondary tumors of the orbit originating from the dura of the sphenoid wing bone. The pathogenesis of intraosseous meningioma remains unclear. A combination of bony hyperostosis and orbital extension are the characteristic imaging features of SOMs which help to differentiate these tumors from sphenoid wing meningiomas and orbital meningiomas [15]. CT examination is a key diagnostic tool in the detection of bony involvement and the hyperostosis of the sphenoid bone, optic canal, anterior clinoid, and orbital fissure. MRI is necessary for the detection of the soft-tissue tumor component and the dural and extradural extension, with the typical pattern of gadolinium enhancement seen on T1-weighted scans.

SOM, despite its benign histopathological characteristics and widespread, carpet-like growth, may cause visual impairment by the invasion of the optic canal and/or by exerting the mass effect in the long clinical course [8]. The latter was the cause of compressive optic neuropathy (CON) in the presented study case. CON is characterized by such clinical findings as decreased CDVA, VF defects, RAPD, abnormal color test results, abnormal VEP, decreased peripapillary RNFL and GCC thicknesses, and a pale optic nerve disc.

The incidence of SOMs is higher in women than in men, with women constituting 85.3% (range 58–100%) of study populations described in papers discussed in this review. The mean age of the cohort was 51.7, and the patients’ ages ranged from 12 to 89 years. A study by Mirone et al. [31], including 71 consecutive patients with SOMs, showed the following age distribution: 0 to 30 years: 1.4%, 30 to 40 years: 15.5%, 40 to 50 years: 16.5%, 50 to 60 years: 45.1%, 60 to 70 years: 14.1%, and 70 years and older: 7.4%.

### 5.1. Preoperative Visual Impairment

The study case manifested painless proptosis, visual impairment, periorbital edema, and headache. Moreover, the mass effect of the tumor in the left orbit resulted in extraocular muscle deformation and eyeball compression with secondary, slightly elevated IOP in the affected eye. Other common symptoms of SOM include cranial nerve deficits, temporal edema, oculomotor deficits, and retrobulbar pain [3,5,6,25,28,30,33]. Periorbital edema was reported to occur in every third patient with SOM [2]. Pain may result from increased intracranial and/or intra-orbital pressures, nerve infiltration by a tumor, and dural involvement. The duration of preoperative symptoms ranged from 1 to 120 months in the study cohort. The average duration of vision loss was 7–21 months [2,5,14,17,26].

According to the reviewed literature, visual impairment defined as a decrease in CDVA was present preoperatively in 684 patients (54.8% of the study cohort). It was shown by Amirjamshidi et al. [22] that a longer symptom duration (47 months) was associated with a higher proportion of patients (74%) complaining of compromised vision.

VF impairment was monitored only in 14 studies, which included the total of 352 patients with SOMs, and 38.3% of this group of patients (n = 135) had demonstrated preoperative VF defects. The pattern of visual field (VF) defects in patients with SOMs included peripheral or central scotomas, generalized depression, and a constricted VF [4,30].

### 5.2. Tumor Surgery and Factors Influencing Postoperative Visual Outcomes

All patients with SOMs included in the reviewed articles underwent surgery. It was confirmed by many authors that the surgical management of SOMs is considerably challenging because of the infiltrative nature of multicompartmental lesions. A complete resection of the tumor is commonly limited when respecting the anatomical limitations as the tumor often infiltrates the superior orbital fissure, cavernous sinus, extraocular muscles, and nerves in the orbit. Therefore, a surgical strategy aims at safe maximal resection rather than aggressive gross total resection [15]. Subtotal resection occurs in 10–80% of the procedures [4,6,11,25,34]. In the study case, postoperative contrast-enhanced MRI showed a tumor residue. Based on the World Health Organization (WHO) classification, postoperative histopathological findings revealed meningothelial meningioma cells (WHO Grade I) with the features of infiltration into the dura, bone structure, and venules. In the reviewed literature, benign WHO Grade I meningiomas constituted the vast majority of the excised tumors. Several Grade II (atypical lesion) cases and no Grade III (anaplastic or malignant tumor) cases were reported.

Surgery improved the visual prognosis in the presented study case. According to the analyzed literature, CDVA improved in 57.0% of patients postoperatively and was stable in 25.9% of patients with preoperative visual impairment due to SOM. Vision continued to deteriorate in 17.1% of the patients following the surgery. Among 123 patients with SOMs who demonstrated preoperative VF defects, VF improvement or normalization was observed in 54.1% and 36.3% of patients, respectively. Further VF deterioration was experienced by 9.6% of patients with preoperative VF defects.

According to Gonen et al. [2], postoperative visual outcomes were significantly related to two preoperative parameters: the severity of visual impairment and the tumor involvement of the optic canal. Mirone et al. [31] demonstrated preoperative CDVA better than 2/10 in 24 (80%) out of 30 patients with improved visual function after surgery. It was confirmed by Menon et al. [15], who also found that better postoperative vision outcomes were observed in patients with a symptom history of less than two years. In turn, Mirone et al. [31] observed the best visual recovery when the duration of ocular symptoms did not exceed six months.

In the reviewed case series, CDVA deteriorated in 17.1% of patients following surgery. It was shown by Najafabadi et al. [14] with regression analysis that poorer preoperative CDVA was the predictor for poorer short-term (three months) and longer-term (at the longest follow-up) postoperative CDVA. Each point of lower preoperative CDVA (measured with the Snellen chart) translated into postoperative CDVA lower by 0.49. In turn, no predictors were identified for short-term postoperative VF outcomes [14]. Nochez et al. [7] demonstrated that severely compromised vision before surgery was a negative predictive factor for long-term VF outcomes.

Oya et al. [28] reported that 2 patients with severe vision loss of 12 months experienced a slight recovery of vision following surgery, but no patient with preoperative blindness noticed any functional visual improvement. Such irreversible poor prognosis for vision recovery was also noted by Saeed et al. [27] and Cannon et al. [33] in six patients, who were already unilaterally blind with no light perception before surgery. Moreover, patients who had only light perception preoperatively were at risk of developing monocular blindness following the surgery, as it was documented in several series [18,26].

Several authors showed that patient characteristics including age, the duration of symptoms, extent of proptosis, or the type of surgery did not contribute to the prediction of the visual outcomes [5,15,28]. According to Oya et al. [28], some patients with significantly compromised visual function before surgery did not experience improvement, despite a relatively shorter duration (six months) of their deficits. In contrast, some patients with mild and long-lasting (three years) visual impairment experienced the normalization of their vision [28]. A series by Terrier et al. [6] showed that 44.9% of patients with preoperative visual deficits improved. However, no correlation was found between the severity of preoperative visual impairment and the postoperative visual outcomes.

Multiple regression analysis adjusted for imaging tumor characteristics revealed that a markedly enlarged sphenoid bone was an independent favorable factor for a better visual outcome after surgery [28]. According to Mariniello et al. [8], the intra-orbital location of the tumor and the pattern of invasion of the optic canal (lateral, medial, or concentric) were significant predictors influencing postoperative visual outcomes. The rate of visual improvement was significantly higher in cases with lateral orbital involvement and the invasion of the optic canal than in those with inferomedial meningiomas (100% vs. 67%). The postoperative visual outcome of the study case is in line with the results obtained by Mariniello [8]. In the presented case, the intra-orbital mass was located on the lateral wall and in the apex of the left orbit. Despite poor preoperative CDVA (Snellen 0.2), a significant visual improvement following tumor resection was observed (Snellen 1.0).

The reviewed series revealed the lowest rate of visual improvement in patients with orbital apex tumors and the concentric invasion of the optic canal [7,8,28]. Other factors that were correlated with the visual outcome included the Simpson grade of surgical resection [36]. The regression analysis conducted by Najafabadi et al. [14] showed that a larger maximum diameter of preoperative hyperostosis and a smaller extent of tumor resection (Simpson grade IV and III) were the predictors for poorer postoperative visual fields. Each additional millimeter of preoperative hyperostosis and each increase in the Simpson grade translated into a postoperative VF decrease by 0.39 and 3.71 dB, respectively [14]. Other authors also confirmed that the cases of SOMs with incomplete resection mainly belonged to the group of orbital apex and diffuse tumors [10] and visual improvement was observed in only 8.3–41.7% of patients in those subpopulations.

Probable reasons for visual decline following surgery include the impairment of the vascular supply to the optic nerve accompanying microsurgical maneuvers and subsequent ischemic optic neuropathy (ION) and/or retinal ischemia, intraoperative stress, or inadvertent injury to the already compromised ON, the elevation of intracranial pressure, cerebral ischemia, and finally, thermal injury during optic canal opening [33]. Posterior ION, related to the hemodynamic disturbances within the pia vessels derived from the branches of the ophthalmic artery, is the most common cause of vision loss following tumor resection [37].

Subtotal tumor resection in the study case provided a full restoration of CDVA, VF, and color vision, as well as a partial recovery of PVEP. Persistently decreased SS-OCT RNFL and GCC in the affected eye were observed through the 78-month observation. Preoperatively, we found significant interocular differences in RNFL thickness in the temporal quadrant and of macular GCC thickness. Park et al. [38] conducted a study in patients with CON in the course of dysthyroid orbitopathy and demonstrated that RNFL thickness could be used as a prognostic factor for visual outcomes before decompression surgery. Patients with greater preoperative inferior peripapillary RNFL thickness tended to have better postoperative CDVA 6 months after orbital wall decompression [38]. The most recent studies on CON have highlighted the role of preoperative nasal GCC thickness as a stronger marker than temporal RNFL thickness in predicting VF improvement after cranial surgery [39].

We also found PVEP latency delay and an increase in PVEP amplitudes. Both endogenic and exogenic determinants might serve as explanations of this phenomenon. Firstly is an ongoing process of axonal degeneration, as it is known that compressive tumors may cause the demyelination and distortion of the nodes of Ranvier [40]. It might correspond with the postoperative PVEP latency delays in the affected eye that were comparable to those in patients affected by optic neuritis. Secondly, as PVEP encompasses responses from the central 15 degrees of VF, an increase in PVEP amplitude, which was observed in the presented case after surgery, might be correlated with CDVA and VF improvement. Therefore, PVEP amplitudes seem to be a better clinical marker to follow the clinical course of CON than PVEP latencies [41]. Thirdly, an exogenic injury of the ON during tumor resection should also be considered as a reason for progressive thinning of ON and inner layers of the retina, which was observed postoperatively in SS-OCT protocols.

Our observation and the available literature sources on unilateral CON showed that VF, OCT, and PVEP examinations should be viewed as complementary methods for providing essential information about the morphological and functional state of the visual pathway in patients with SOMs. PVEP revealed visual impairment in our patient who had no subjective complaints about visual acuity postoperatively. To the best of our knowledge, this is the first case-report study of the use of OCT RNFL/GCC and PVEP to detect structural and functional changes in the visual pathway in the follow-up of patients with SOMs. The potential of these diagnostic tools to predict postoperative visual prognosis needs to be established in future studies.

Conclusions reported in the published literature suggested that it might be beneficial to operate on patients with early visual impairment or limited hyperostosis to prevent further visual deterioration. Moreover, performing the surgery early in the course of the disease when the ON is less vulnerable (because of less severe compression) increases the safety profile of the intervention [14]. Decisions about the surgery should be made on an individual basis after weighing the benefits of surgery and the risk of complications. Surgery itself may also impose a risk of further visual deterioration in patients with very low preoperative vision, in very old patients, and in patients with severe comorbidities [6,13,14].

The mean follow-up of the analyzed studies was 57.7 months and ranged from 6 [24] to 136 months [22]. The rate of long-term visual deterioration is known to be related to the length of follow-up and ranged from 0% to 39% in studies with the mean follow-up of 12 months [2] and 6.8 years [18], respectively.

### 5.3. Rate and Risk Factors of Tumor Recurrence

Several MRI examinations were performed in the study case to assess possible tumor recurrence and increases in the size of the residue. We found no tumor recurrence at the follow-up of 78 months and CDVA of the left eye was 1.0 (20/20). The evolution of residual tumors varies from complete stability for years, through very slow regrowth observed in most cases, to a rapid increase in the size of the residue. Marinelli et al. [10] showed that the extent of tumor resection constituted one of the most important predictive risk factors of recurrence for SOMs. The invasion of the whole orbit, the involvement of the orbital apex, and superior orbital fissure are the risk factors of tumor recurrence [10].

The mean rate of recurrence in the reviewed series was 24.3% (range 0–59%). A recurrence rate of 5% was reported in a series of patients, 83% of whom achieved total tumor removal [1,31]. When the rate of complete resection was as low as 31%, the rate of tumor recurrence was 29% [20]. The rate of recurrence, similar to the rate of long-term visual deterioration, depends on the length of follow-up. It ranged from 6% to 14% in studies with the follow-up shorter than 3 years [1,12] and from 33% to 59% in studies with the follow-up of 56–89 months [7,15,33,34]. The recurrence rate in the study with the longest follow-up (5–28 years, mean 136 months) was 37.5% in this review [10]. According to Talacchi et al. [23] and Terrier et al. [5], long-term follow-up was recommended as majority of recurrences appeared six years after the surgery. The mean time to recurrence in the reviewed series was 57.7 months (range: 10–108 months). The most common symptoms related to the recurrence of SOM included decreased visual acuity or blurring, progressive proptosis, and diplopia. Marcus et al. [26] recommended to extend follow-up periods to over 20 years postoperatively.

There are some limitations that need to be addressed. Firstly, the study cohort included both patients with newly diagnosed SOMs as well as patients presenting with the recurrences of previously resected tumors. Secondly, visual outcomes following surgery were evaluated at different postoperative time points and both short-term (six months) and long-term (reporting at last examination) visual outcomes were included in the reviewed studies. Thirdly, a unimodal approach, based on CDVA, was applied in majority of studies for the measurement and monitoring of ON function. This might produce the risk of the underestimation of ON dysfunction detection in study populations both pre- and post-operatively. It was demonstrated by Young [18] that using only CDVA as a measure of ON function instead of a multimodal approach could have resulted in missing almost 30% of cases with ON dysfunction.

## 6. Conclusions

To sum up:-SOMs are mostly found in women who are around 50 years of age and have a high rate of visual impairment (mean 56.0%).-A growing body of literature published in the last two decades suggests that early surgical treatment of symptomatic SOMs improves visual outcomes, giving the less vulnerable optic nerve a better chance to recover its function.-Favorable factors for better visual outcomes after surgery included better preoperative CDVA, a symptom history of less than 2 years, and lateral orbit involvement. Severely compromised visual function before surgery, orbital apex tumors, and the concentric invasion of the optic canal, as well as a lower extent of tumor resection (Simpson grades IV and III), were the predictors for a poorer postoperative visual prognosis.-SOMs pose a surgical challenge and complete surgical resection is often not possible. Patients with SOMs required a long-term follow-up because of the delayed high rate of recurrence.

## Figures and Tables

**Figure 1 jcm-12-00074-f001:**
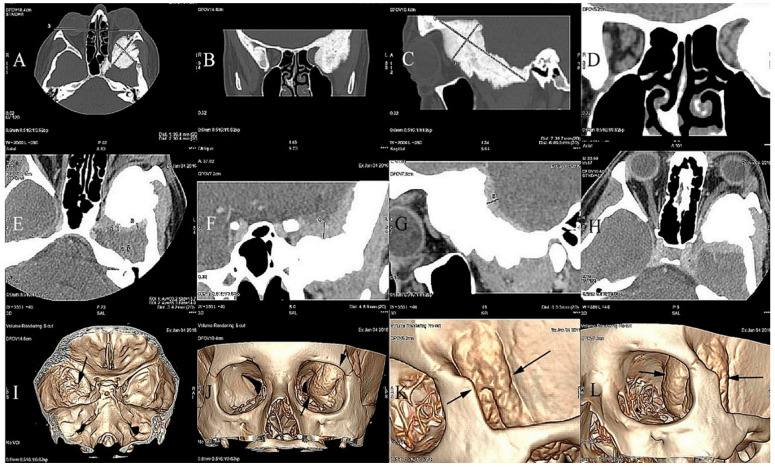
Craniofacial CT scan demonstrating an osteoblastic tumor measuring 36.4 × 30.4 × 89.5 mm that is invading the left greater wing of the sphenoid bone. (**A**) Exophthalmos of the left eye determined with reference to the inter-zygomatic line. Spiculated periosteal reactions (**A**–**C**). Orbital apex narrowing exerting pressure on the oculomotor muscles and the optic nerve (**D**). Contrast enhancement of the dura mater of the sphenoid bone up to +99 HU caused by tumor infiltration in the axial, coronal, and sagittal sections (**E**–**G**). Compression of the superior rectus muscle and narrowing of the optic nerve canal (**H**). Deformities of the left orbit caused by the meningioma (**I**–**L**) iIn 3D-CT reconstructions. The borders of the tumor are marked with black arrows.

**Figure 2 jcm-12-00074-f002:**
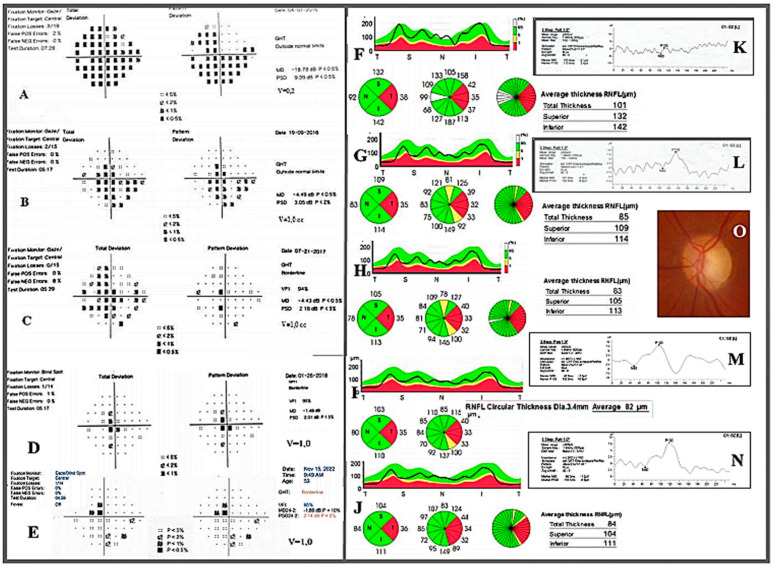
(**A**–**E**) Humphrey 24–2 standard automated perimetry of the left eye. (**F**–**J**) RNFL SS–OCT examination of the left eye. (**K**–**N**) PVEP responses obtained from the left eye, a week before the surgery (**A**,**F**,**K**), 5 months after the surgery (**B**,**G**,**L**), 15 months after the surgery (**C**,**H**), 22 months after the surgery (**D**,**I**,**M**), and 78 months after the surgery (**E**,**J**,**N**). (**O**) Pale optic nerve disc of the left eye.

**Figure 3 jcm-12-00074-f003:**
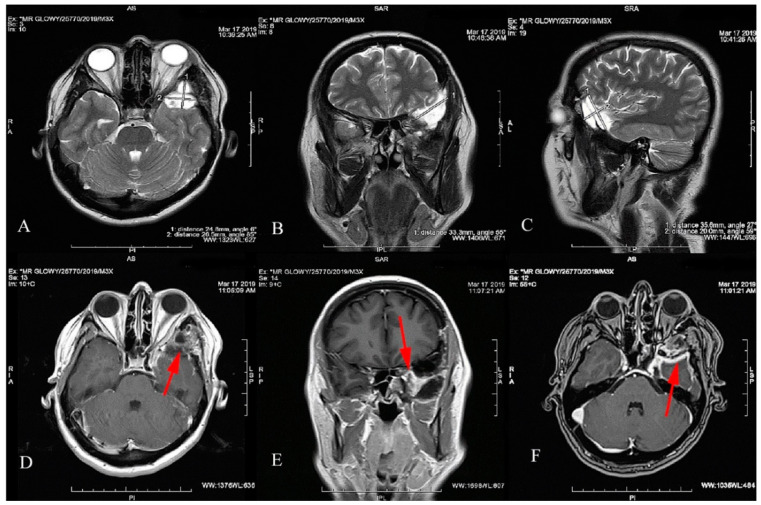
Postoperative MRI T2-weighted images with the SOM resection focus measuring 33.3 × 20 × 35.6 mm (**A**–**C**). The residual mass of the tumor shown in the T1-weighted images (**D**–**F**). The residue compresses the left temporal pole and is enhanced after gadolinium administration. The location of the tumor residue is marked with a red arrow.

**Figure 4 jcm-12-00074-f004:**
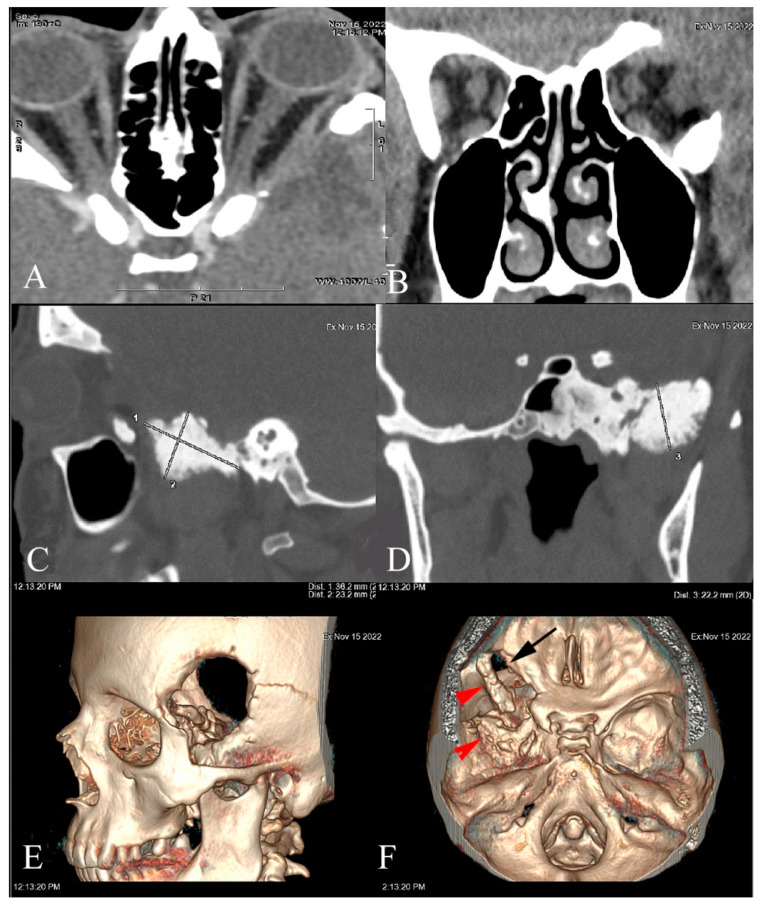
CT scan performed 78 months after SOM resection with orbital lateral wall removal and decompression (**A**,**B**). Incomplete resection, residue size 36.2 × 23.2 × 22.2 mm in the left sphenoid outside the orbit (**C**,**D**). Left craniotomy on 3D reconstructions (**E**), and partial resection of the sphenoid (black arrow) and SOM (red arrow) (**F**).

**Figure 5 jcm-12-00074-f005:**
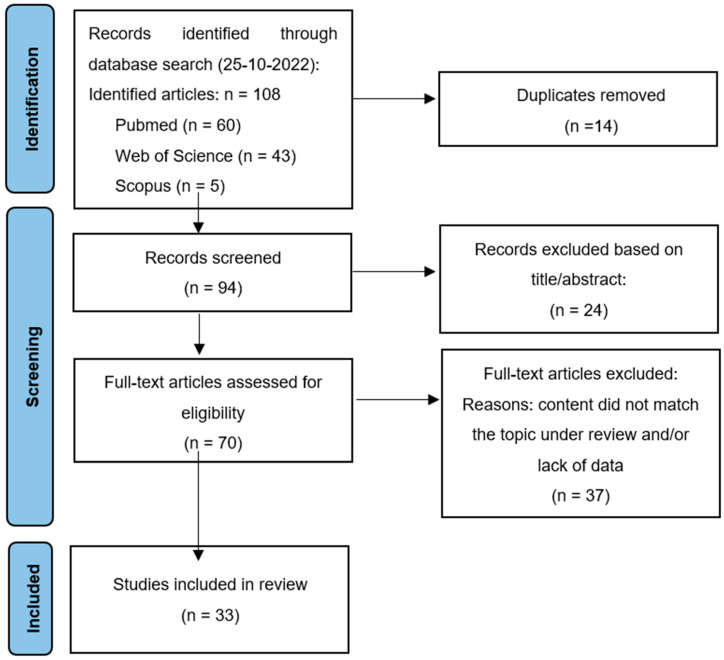
PRISMA diagram of study selection process for this review.

**Table 1 jcm-12-00074-t001:** Characteristics of the studies included in the review.

Study	Study Period	Study Size	Age Mean (Range)	Female %	Visual Impairment		Visual Improvement *		Visual Deterioration *		Follow-Up Mean (Months)	Tumor Recurrence %
					VA n (%)	VF n (%)	VA n (%)	VF n (%)	VA n (%)	VF n (%)		
Mariniello et al., 2022 [10]	1990–2014	80	47 (26–75)	82.5	47 (59)	NS	24 (51)	NS	7 (15)	NS	136	37.5
Dos Santos et al., 2022 [11]	2008–2018	40	49.5	87.5	26 (65)	NS	13 (41)	NS	7 (22)	NS	39	25
Locatelli et al., 2022 [12]	2011–2021	35	57 (38–80)	77	11 (32)	6 (17)	(64)	2 (33)	0 (0)	0 (0)	31.5	14
Dalle Ore et al., 2021 [13]	NS	54	52 (30–79)	83	28 (52)	28 (52)	12 (43)	12 (43)	1 (3.5)	0 (0)	31	22
Najafabadi et al., 2021 [14]	2015–2019	19	47 (45–50)	95	10 (53)	8 (42)	8 (80)	8 (100)	2 (20)	0 (0)	29	10.5
Menon et al., 2021 [15]	10 years	17	51 (17–72)	76	14 (82)	NS	4 (29)	NS	2 (14)	NS	56	59
Samadian et al., 2020 [16]	2007–2017	57	48 (22–76)	93	16 (28)	13 (23)	7 (44)	NS	1 (6)	NS	46	17
Pace et al., 2020 [17]	1996–2017	20	56 (19–89)	80	11 (55)	15 (75)	9 (82)	11 (73)	3 (20)	1 (7)	47	20
Young et al., 2019 [18]	2000–2017	24	49.5	92	17 (71)	16 (67)	12 (71)	7 (44)	5 (29)	5 (31)	82	33
Nagahama et al., 2019 [19]	1996–2017	12	49 (20–71)	58	3 (25)	NS	NS	NS	2 (66)	NS	74	33
Terrier et al., 2018 [5]	20 years	130	51 (28–74)	91,5	49 (38)	NS	22 (45)	NS	8 (16)	NS	77	25
Gonen et al., 2018 [2]	2005–2014	27	53 (27–78)	89	10 (37)	4 (15)	8 (80)	NS	0 (0)	NS	41	11
Freeman et al., 2017 [3]	2000–2016	25	51 (39–71)	92	19 (76)	NS	4 (21)	NS	1 (5)	NS	45	48
Leroy et al., 2016 [20]	2001–2006	70	52 (21–80)	91	27 (39)	NS	10 (37)	NS	6 (22)	NS	57	29
Bowers et al., 2016 [21]	2002–2015	33	52 (12–76)	73	17 (51.5)	12 (36)	15 (68)	NS	0 (0)	NS	54	6
Amirjamshidi et al., 2015 [22]	1979–2013	88	46 (12–70)	74	65 (74)	NS	39 (60)	NS	13 (20)	NS	135	22
Talacchi et al., 2014 [23]	1992–2012	47	57 (21–77)	55,5	24 (51)	6 (13)	10 (42)	3 (50)	3 (12.5)	1 (17)	52	30
Berhouma et al., 2014 [24]	2012–2014	4	58 (49–67)	75	4 (100)	3 (75)	3 (75)	1 (15)	1 (15)	0 (0)	6	NS
Boari et al., 2013 [25]	2000–2010	40	53	88	35 (88)	NS	27 (67)	NS	NS	NS	73	10
Marcus et al., 2013 [26]	2004–2012	19	44 (26–64)	90	11 (58)	1 (6)	10 (91)	1 (100)	1 (9)	0	60	0
Nochez et al., 2012 [7]	1986–2006	37	50 (33–76)	92	22 (54.5)	NS	12 (71)	NS	5 (23)	NS	89	42.5
Saeed et al., 2011 [27]	1980–2006	66	46 (26–68)	92	51 (77)	51 (77)	20 (39)	NS	6 (12.5)	NS	102	17
Oya et al., 2011 [28]	1994–2009	39	48 (33–68)	87	21 (54)	NS	14 (67)	NS	0 (0)	NS	41	18
Luetjens et al., 2011 [29]	NS	3	62 (49–70)	100	3 (100)	3 (100)	3 (50)	3 (50)	0 (0)	0 (0)	28	NS
Honig et al., 2010 [30]	2001–2006	30	54 (25–74)	73	22 (73)	12 (40)	15 (68)	NS	2 (9)	NS	34	27
Mirone et al., 2009 [31]	1986–2006	71	53 (12–79)	87	41 (58)	27 (38)	30 (73)	NS	3 (7)	NS	77	5
Heufelder et al., 2009 [32]	1997–2006	21	61 (47–81)	95	10 (48)	7 (33)	2 (20)	2 (29)	2 (20)	1 (14)	66	33
Cannon et al., 2003 [33]	2000–2007	12	51 (34–64)	92	5 (42)	10 (83)	2 (40)	1 (10)	3 (60)	3 (30)	31	33
Ringel et al., 2007 [6]	1983–2003	63	51 (21–77)	85	28 (47)	20 (32)	18 (64)	11 (55)	2 (7)	2 (10)	54	39
Bikmaz et al., 2007 [1]	1994–2004	17	52 (36–70)	88	10 (59)	NS	7 (70)	NS	0 (0)	NS	36	6
Shrivastava et al., 2005 [4]	1991–2003	25	51 (22–76)	88	20 (80)	9 (36)	7 (39)	8 (89)	0 (0)	0 (0)	60	8
Sandalcioglu et al., 2005 [34]	1998–2002	16	53 (37–76)	94	7 (44)	NS	6 (86)	NS	1 (14)	NS	68	56
De Jesus et al., 2001 [35]	1990–1997	6	51 (39–64)	100	3 (50)	3 (50)	3 (100)	3 (100)	0 (0)	0 (0)	48	17

VA: visual acuity; VF: visual field; NS: not specified. * The percentage of patients with visual improvement or deterioration refers to the group of patients with preoperative visual impairment.

## Data Availability

The data that support the findings of this study are available upon request from the corresponding author.

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
