# Peer review of "Spheno-Orbital Meningioma and Vision Impairment—Case Report and Review of the Literature"

_jcm, 2022, doi:10.3390/jcm12010074_

Round 1
Reviewer 1 Report
Authors reported a case with Spheno-orbital meningioma (SOM), a very rare subtype of meningioma, and reviewed for SOM.
The writing often lacks clarity and sharpness, and several sections are poorly organized or do not flow with the rest of the paper. The main problem with the paper is the description of the results and discussion of their significance.
Author Response
Point 1:
Authors reported a case with Spheno-orbital meningioma (SOM), a very rare subtype of meningioma, and reviewed for SOM. The writing often lacks clarity and sharpness, and several sections are poorly organized or do not flow with the rest of the paper. The main problem with the paper is the description of the results and discussion of their significance.
Response 1:
We would like to thank Reviewer 2 for the comment. The section of Results has been re-organized and the following 3 additions have been made:
- The description of the data included in Table 1. The following statement has been added on page 7, lines201-203:
“They include the demographic features of SOMs as well as data on preoperative visual impairment, postoperative visual outcomes, the length of follow-up and the rate of tumor recurrence.”
- The description of the extent of tumor resection according to the Simpson grading scale. The following paragraph has been added on page 8, lines 223-231:
“The extent of tumor resection was categorized in the reviewed literature according to the Simpson grading scale [36] as total macroscopic (TTR, Simpson grade I), near-total (NTR, Simpson grade II), subtotal (STR, Simpson grade III) and partial tumor resection (PTR, Simpson grade IV). TTR and NTR were considered when surgery left no visible residual tumor on the follow-up MRI and CT scans, whereas STR and PTR were defined as a residual tumor that was left within the cavernous sinus or intradurally, respectively. Some authors used the term “gross total resection” (GTR) as equivalent to the terms TTR (Simpson grade I) and NTR (Simpson grade II) and others see GTR as equivalent to TTR only.”
- The main characteristics of the tumor surgery and the postoperative outcomes in the reviewed case-series (on page 8-13, lines 231-441 of the manuscript). The description of the results will increase, in our opinion, the clarity of this section and will provide sufficient data to be interpreted in the Discussion section.
For better clarity of the Discussion section, the main text of this part of the manuscript has been divided and the following subtitles have been inserted: “Preoperative visual impairment” on page 14, “Tumor surgery and factors influencing postoperative visual outcomes” on page 14 and “Rate and risk factors of tumor recurrence” on page 17.
Point 2:
English language and style are fine/minor spell check required.
Response 2:
The manuscript underwent extensive English revisions by a Polish-English translator specializing in written medical translation.
Sincerely,
Joanna Wierzbowska

Reviewer 2 Report
This is a very thorough, detailed case report that discusses a less common cause of compressive optic neuropathy. However, the conclusion that surgical decompression is the next step, and that vision can improve to some extent after decompression even when there is a degree of permanent damage/ optic atrophy (which was demonstrated by the RNFL and GCC thinning) is not particularly surprising. The review of the literature is similarly extensive and thorough but conclusions (such as that longer duration of compression and more severe vision loss at presentation leads to less improvement after surgery) and also not surprising. More helpful conclusions were drawn about impact of tumor position within the orbit on the visual prognosis, that these data confirm that surgery improves visual prognosis, and the discussion of documented complications of surgery. On Page 2, Line 77 and Page 3, Line 110, would be important to now how many Ishihara color plates were tested (7 out of how many?, 12 out of how many?). On Page 10, Line 329-330, the authors mention that "neither OCT RNFL nor GCC improvement were observed, following surgery." This statement should be reworked, as obviously RNFL and GCC thinning would not be expected to be reversible, and thus this statement is strange.Author Response
We would like to thank Reviewer 2 for the comments.
Point 1: This is a very thorough, detailed case report that discusses a less common cause of compressive optic neuropathy. However, the conclusion that surgical decompression is the next step, and that vision can improve to some extent after decompression even when there is a degree of permanent damage/ optic atrophy (which was demonstrated by the RNFL and GCC thinning) is not particularly surprising.The review of the literature is similarly extensive and thorough but conclusions (such as that longer duration of compression and more severe vision loss at presentation leads to less improvement after surgery) and also not surprising. More helpful conclusions were drawn about impact of tumor position within the orbit on the visual prognosis, that these data confirm that surgery improves visual prognosis, and the discussion of documented complications of surgery.
Response 1:
We do agree with the Reviewer that surgery is nowadays the first line of the management of SOMs. In the past, the surgical removal of SOMs had been considered as impossible for a long time (or as a last resort), as the results of operative intervention were disappointing [1, 2]. The resection of SOM was first recommended by Derome in the 70's of the XX century.
Since then, the surgery for SOMs has been the subject of the opened discussion among neurosurgeons in terms of surgical approach, the extent of resection of hyperostic bone as well as the necessity and strategy of craniofacial, and orbital reconstruction. It should also be emphasized, on the basis of the reviewed literature, that the surgical management of SOMs has its specific characteristics: 1/ complete resection is often impossible and the surgeon often faces the determination whether the risk of further tumor removal is greater than the benefit of total resection; 2/ the recurrence rate of SOMs is higher than that of meningiomas in other locations.
We also do agreee with the Reviewer that the recovery of vision following decompression for SOMs may and does occur even when there is a degree of permanent damage or atrophy of the optic nerve. However, postoperative visual loss is a very rare, but well known complication of the surgery for SOMs, which was also discussed in our review. Transient or permanent visual acuity loss following surgery for SOMs was reported by some authors in 20% and 7-12% of cases, respectively [2,3]. To the best of our knowledge, the time course of visual field recovery and changes of RNFL and retinal ganglion cells measured with OCT after the decompression for SOMs has not been evaluated so far.
Finally, we also do agree with the Reviewer that longer compression and more severe vision loss at presentation leads to less improvement after the surgery. This statement may be not a surprising one. However, several authors included in our review showed that there were no correlations between the duration of symptoms [4-6] as well as the severity of preoperative visual impairment [7] and the postoperative visual outcomes.
- S, Pellerin P., Dhellemmes P et al. Strategy of Craniofacial Reconstruction after Resection of Spheno-orbital “en Plaque” Meningiomas. Plast Reconstr. Surg. 1997;100:1113—1120.
- Honeybul S, Neil-Dwyer G, Lang DA, Evans BT, Ellison DW. Sphenoid Wing Meningioma en Plaque: A Clinical Review Acta Neurochir (Wien) (2001) 143: 749±758
- Terrier LM, Bernard F, Fournier HD et al. Spheno-Orbital Meningiomas Surgery: Multicenter Management Study for Complex Extensive Tumors. World Neurosurg. 2018;112:e145-e156.
- Terrier, L.M.; Fournier, H.D.; Morandi, X.; Velut, S.; Hénaux,L.; Amelot, A.; François, P. Spheno-Orbital Meningiomas Surgery: Multicenter Management Study for Complex Extensive Tumors. World Neurosurg 2018,112,e145-e156.doi:10.1016/j.wneu.2017.12.182.
- Menon, S.;Sandesh, O.; Anand, D.; Menon, G. Spheno-Orbital Meningiomas: Optimizing Visual Outcome. J Neurosci Rural Pract 2020,11(3),385-394. doi: 10.1055/s-0040-1709270.
- Oya, S.; Sade, B.; Lee, J.H.Sphenoorbital meningioma: surgical technique and outcome. J Neurosurg. 2011,114(5),1241-1249. doi: 10.3171/2010.10.JNS101128.
- Ringel, F.; Cedzich, C.; Schramm, J. Microsurgical technique and results of a series of 63 spheno-orbital meningiomas. 2007,60(4 Suppl 2),214-221; discussion 221-2. doi: 10.1227/01.NEU.0000255415.47937.1A.
Point 2: On Page 2, Line 77 and Page 3, Line 110, would be important to now how many Ishihara color plates were tested (7 out of how many?, 12 out of how many?).
Response 2:
We would like to thank the Reviewer for noticing the lack of the information how many Ishihara color plates were tested in an original article. Color vision was tested with the Ishihara pseudoisochromatic plates and the results were recorded as the percentage of plates correct out of 14. The descripion of the pre- and postoperative results of the color vison test were corrected in the manuscript as follows:
- On page 2, lines 79-80: “ Relative afferent pupillary defect (RAPD) was present in the LE and the color tests score (by the Ishihara color plate test) was decreased to 7/14.” and
- On page 3, line 122: “ Color vision score improved from 7/14 to 12/14.”
Point 3: On Page 10, Line 329-330, the authors mention that "neither OCT RNFL nor GCC improvement were observed, following surgery." This statement should be reworked, as obviously RNFL and GCC thinning would not be expected to be reversible, and thus this statement is strange.
Response 3:
Thank you for this comment. The measurement of RNFL and GCC (or GCIPL) is an objective method of quantifying the damage to the visual pathways. It was shown that both markers correlated with perimetry in compression syndrome. Following decompression, the vast majority of patients show persistent RNFL and GCC/GCIPL loss after visual recovery due to axonal and RGCs injury. However, it was also documented that some patients might experience improvement in some sectors of RNFL and/or GCC/GCIPL postoperatively [1].
The mechanism of visual recovery includes the initial removal of the conduction block due to the compression, secondary remyelination, followed by the restoration of axoplasmic flow. The immediate recovery of visual function results from the removal of the conduction block due to compression and the restoration of signal conduction. The delayed restoration of RGCs may take place when active remyelination and axoplasmic flow restoration occurred, resulting in additional improvements of visual function [2]. The time difference between changes in the visual field and RGCs as well as between axon injury and after the loss of RGCs was confirmed in animal studies. Their findings suggest that damage to RGCs does not occur via a direct process, but is rather the consequence of retrograde degeneration and, thus, takes some time to occur. It was shown that, when the site of axonal injury was further from the globe and the injury was mild, the latency to the onset of ganglion cell loss was longer and fewer ganglion cells were lost.The findings of Moon et al. study [3] suggest that changes in the visual field and RGCs in compression syndrome and after decompression show a similar time course aspect, but occur at different time points. Therefore, in patients with early and/or mild compression causing only a conduction block, more complete functional and structural recovery (resulting in GCC or RNFL thickening) is possible. In turn, significant preoperative reduction in RNFL and GCC/GCIPL thicknesses indicate that compression resulted in more prolonged degeneration with subsequent axonal injury and RGC atrophy.
We do agree with the Reviewer that the statement “neither OCT RNFL nor GCC improvement were observed, following the surgery." should be reworked. In the presented case, RNFL and GCC thicknesses measured with SS-OCT were significantly decreased in the affected eye compared to the opposite eye and the temporal part of the optic disc was pale. To the best of our knowledge, the changes of the RNFL and GCC/GCIPL thicknesses in compression syndrome and after decompression in the course of SOMs have not been evaluated so far. On the basis of the literature on compressive optic neuropathy in the course of pituitary adenoma with chiasmal compression or dysthyroid orbitopathy we may assume that the possibility of GCC or RNFL recovery following decompression was rather impossible in our clinical case. The statement on page 16, lines 587-588 was replaced by a new one. Now it says: “Persistently decreased SS-OCT RNFL and GCC in the affected eye were observed through the 78-month observation”.
Ref.
- Agarwal R, Jain VK, Singh S. et al. Segmented retinal analysis in pituitary adenoma with chiasmal compression: A prospective comparative study Indian J Ophthalmol 2021;69(9):2378-2384.doi: 10.4103/ijo.IJO_2086_20.
- Micieli JA, Newman NJ, Biousse V. The role of optical coherence tomography in the evaluation of compressive optic neuropathies. Curr Opin Neurol 2019;32:115-23.
- Moon CH, Hwang SC, Ohn YH, Park TK. The time course of visual field recovery and changes of retinal ganglion cells after optic chiasmal decompression. Invest Ophthalmol Vis Sci. 2011; 10;52(11):7966-73. doi: 10.1167/iovs.11-7450.
Point 4: Extensive editing of English language and style required.
Response 4: The manuscript underwent extensive English revisions by a Polish-English translator specializing in written medical translation.
Sincerely,
Joanna Wierzbowska

Reviewer 3 Report
Congratulations. Although it is only a case report a good review of literature has been made. Therere are previous literature reviews, but the add of multimodal evaluation is the key of current evaluation of any optic nerve disease. It is strange that no other articles are published about multimodal evaluation.

Author Response
Point 1: Congratulations. Although it is only a case report a good review of literature has been made. Therere are previous literature reviews, but the add of multimodal evaluation is the key of current evaluation of any optic nerve disease. It is strange that no other articles are published about multimodal evaluation.
Response 1:
We would like to thank the Reviewer for the comments.To the best of our knowledge, no studies are available that address a multimodal approach to evaluate both the function and structure of the optic nerve, including vision acuity, visual field, RAPD, color test, visual evoked potential, peripapillary RNFL and GCC/GCIPL thicknesses in the course of spheno-orbital meningiomas. The time course of visual field recovery and changes of RNFL and retinal ganglion cells measured with OCT after the decompression for SOMs has not been evaluated so far.
Point 2: English language and style are fine/minor spell check required.
Response 2:
The manuscript underwent extensive English revisions by a Polish-English translator specializing in written medical translation.
Sincerely,
Joanna Wierzbowska

Round 2
Reviewer 1 Report
Authors reported a case with Spheno-orbital meningioma (SOM), a very rare subtype of meningioma, and reviewed for SOM. The authors have restructured the results section to provide a very clear context. By restructuring the Results section, the Discussion section also became easier to understand. This paper is a valuable summary of the very rare disease SOM.